

# Integrating humanoid robots with human musicians for synchronized musical performances

MengCheng Lau[1], John Anderson[2] and Jacky Baltes[3]

[1] Bharti School of Engineering and Computer Science, Laurentian University, Sudbury, Ontario, Canada
[2] Department of Computer Science, University of Manitoba, Winnipeg, Manitoba, Canada
[3] Department of Electrical Engineering, National Taiwan Normal University, Taipei, Taiwan

## ABSTRACT

Entertainment robotics has garnered significant attention in recent years, with researchers focusing on developing robots capable of performing a variety of tasks, including magic, drawing, dancing, and music. This article presents our research on forming a musical band that includes both humanoid robots and human musicians, with the goal of achieving natural synchronization and collaboration during musical performances. We utilized two of our humanoid robots for this project: Polaris, a mid-sized humanoid robot, as the drummer, and Oscar, a Robotis-OP3 humanoid robot, as the keyboardist. The technical implementation incorporated essential components such as visual servoing, human-robot interaction, and Robot Operating System (ROS), enabling seamless communication and coordination between the humanoid robots and the human musicians. The success of this collaborative effort can be both seen and heard through the following YouTube link: https://youtu.be/pFOyt1KKCfY?feature=shared.

## INTRODUCTION

Robots have traditionally faced challenges when it comes to performing creative tasks, particularly in the domain of music. For instance, *Nikolaidis & Weinberg (2010)* introduced an approach that uses the iPhone as an instrument to enable interactive musical collaboration between humans and a marimba-playing robot, *Shimon*, creating improvisatory exchanges influenced by jazz masters and allowing users to control and inspire *Shimon*'s musical responses. However, there have been notable advancements in this research area. German artist Frank Barnes introduced *Compressorhead*, the first robot band capable of performing live in front of an audience. These robots played real electric and acoustic instruments controlled *via* a musical instrument digital interface (MIDI) sequencer, as shown in *Davies & Crosby (2016)*. Subsequently, the Yuri Suzuki Design Studio presented Z-Machinese, a Japanese robot band comprising a 78-finger guitarist, a 22-arm drummer, and a keyboardist using lasers to trigger notes as featured in *McKenzie (2014)*. Another remarkable example is the improved version of *Shimon*, a smarter robot musician introduced by *Bretan & Weinberg (2016)* from the Georgia Tech Center for Music Technology. *Shimon* played marimba alongside a human musician and used deep

Corresponding author
MengCheng Lau,
mclau@laurentian.ca

neural networks to learn and compose its own pieces. For several decades, Waseda University has developed anthropomorphic robots, including WABOT-2 a keyboardist, WF-4RIV a flutist and WAS-1 a saxophonist, capable of playing musical instruments and interacting in performances, respectively, by *Solis et al. (2009)*, *Kato et al. (1987)*, *Sugano & Kato (1987)*, *Solis et al. (2008)*, *Petersen et al. (2009)*. Another example is the HRP-2 humanoid robot (*Kaneko et al., 2004*), which has been used for drum-playing and theremin-playing tasks (*Mizumoto et al., 2010*; *Konno et al., 2007*). Their research aims to create robot orchestras, enhancing musical expression and understanding human communication in music, with initial experiments enabling duet performances using MIDI data. Recently, *Karbasi et al. (2022)* presented a drum robot with two degrees of freedom and a quasi direct-drive servo motor, and explored its ability to perform both single and double-stroke drum rolls on a snare drum, providing a foundation for the development of algorithms enabling the robot to learn musical patterns. Some research such as *Crick, Munz & Scassellati (2006)*, *Hoffman & Weinberg (2011)*, *Chida, Okada & Amemiya (2014)*, *Chakraborty & Timoney (2020)* have focused on improving robots' ability to synchronize with human musicians. These research emphasized the importance of sensory integration and prediction in achieving synchronized performances. Additionally, image-based recognition systems have enhanced robots' ability to interpret sheet music and identify instruments visually, combined with audio-based synchronization and visual cue following, has led to more sophisticated multi-modal integration approaches. These developments are crucial for enabling robots to interact more naturally and responsively in musical settings with human partners as shown by *Lim, Ogata & Okuno (2010)*. While these robots demonstrated impressive musical abilities, they relied on intricate mechanical designs with rigid setups, lacking the versatility and robustness of intelligent humanoid robots.

The primary motivation for our research is to explore the feasibility of creating a rock band that does not rely on intricate mechanical designs. By utilizing our existing humanoid robots, which are capable of performing various instruments together, we aim to develop a system that is both robust and versatile. This approach will enable the robots to play along with human musicians seamlessly. Our goal is to demonstrate that sophisticated musical performances can be achieved with humanoid robots, leveraging their existing capabilities and ensuring they can interact and synchronize effectively with human band members. Over the past few years, we have achieved remarkable success in the Humanoid Application Challenge (HAC) competition by winning 2nd place in 2018 and 1st place in 2017 for the robot magic theme, respectively, in *Morris et al. (2018a, 2018b)*. HAC was an international robotics competition organized as part of the IEEE International Conference on Intelligent Robots and Systems (IROS). The goal of the HAC is to showcase innovative applications of humanoid robots in real-world scenarios, promoting research in robotics that combines advanced algorithms, machine learning, and mechanical design. Each year, the challenge focuses on a specific theme, encouraging teams to push the boundaries of humanoid robot performance in areas such as entertainment. In 2019, the robot music theme was introduced and we responded to the challenge by developing a humanoid robot band that competed in the Humanoid Application Challenge competition, achieving

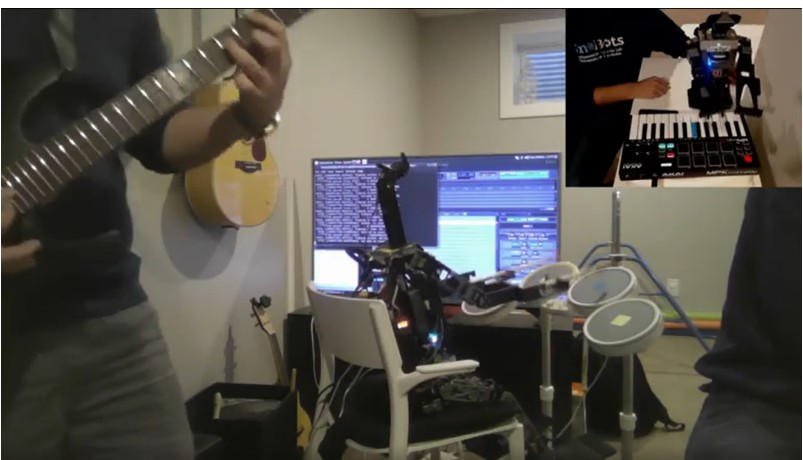

**Figure 1** Performance at IROS Humanoid Application Challenge 2020.

distinguished 2nd place in both 2019 and 2020 (as shown in Fig. 1), demonstrating the level of sophistication and innovation required to achieve recognition in this highly competitive event. By expanding our research into robot music, we introduced two humanoid musicians. First, there is Oscar, our keyboardist, who is a Robotis-OP3 proficient in playing the MIDI Keyboard. Second, we have Polaris, a custom-built mid-sized humanoid drummer, capable of playing Wii Rock Band's drums.

In this work, we present a unified ROS framework that integrates both Polaris and Oscar, allowing for a seamless collaboration between humanoid robots and human musicians. The novelty of our system lies in the unified framework that enables both robots to perform synchronously using a centralized control architecture, which significantly reduces latency and ensures real-time communication. Unlike previous robotic music systems, such as Shimon, which relied on complex mechanical designs and predefined MIDI sequences, our framework introduces a dynamic beats per minute (BPM) synchronization mechanism. This synchronization ensures precise timing between the robots during performances, allowing Polaris and Oscar to adjust their tempo in real-time based on User Datagram Protocol (UDP) messages. This approach not only enhances the performance quality but also introduces flexibility in adapting to different musical pieces, overcoming the latency issues encountered in earlier robotic music systems. By leveraging this unified framework, we aim to provide a more cohesive and versatile platform for humanoid robots in musical performances. In this study, we aim to answer the following research questions: (1) Can humanoid robots synchronize their performances with each other and with human musicians in real-time musical performances? (2) Can the UDP-based synchronization mechanism minimize timing drifts during performances to ensure seamless integration between robots and human performers?

This article provides a comprehensive overview, starting with the roles our robots play, followed by the hardware and software implementation. To facilitate cooperation between these humanoid robots, we implemented a unified ROS (Robot Operating System) framework, ensuring a cohesive software environment. Then in the experimental

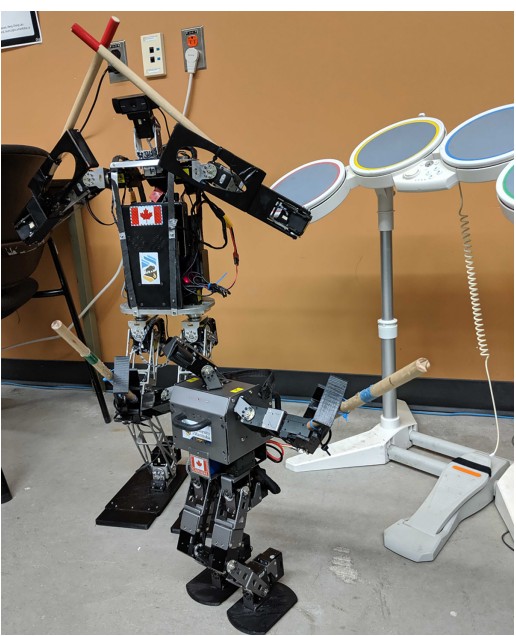

**Figure 2  Snobots' humanoid musicians: polaris (left) and Oscar/Robotis-OP3 (right).**

discussions, we will discuss our findings and insights, underscoring the potential of humanoid robots in creating captivating and collaborative musical experiences. By merging the creative and technological domains, our work not only showcases the potential of entertainment robotics but also highlights the significance of humanoid robots in creating seamless and engaging interactions with humans in musical performances.

## HUMANOID MUSICIANS: POLARIS AND OSCAR

The humanoid robot band consists of two humanoid robots, Polaris and Oscar, as shown in Fig. 2. On the left side of Fig. 2 stands Polaris, a custom-built mid-sized humanoid robot that stands 92 cm tall and weighs 7.5 Kg, with 20 DOFs (Degrees of Freedom). All the servos use the transistor-transistor logic (TTL) serial communication protocol, with three pins that share one line for sending and receiving data. Each hip can rotate in the lateral, frontal, and transversal planes. There are 6 MX-106 servos in each leg (hip transversal, hip lateral, hip frontal, knee lateral, ankle frontal, and ankle lateral). The two arms rotate in the lateral and frontal planes, and each arm has one MX-106 and two MX-64 servos (shoulder lateral, shoulder frontal, and elbow lateral). The neck is made up of 2 MX-28 servos and rotates in the lateral and transversal planes, moving an attached webcam. An Intel dual-core i5-based NUC (8 GB DDR4 SODIMMs 2,133 MHz and 256 GB M.2 SSD) is used for computation. On the right of Fig. 2 is our keyboard player Oscar, an Robotis-OP3 humanoid by *Robotis (2024a)*, standing 51 cm tall and weighing 3.5 Kg. Oscar features 20 DOFs (two DOFs on the neck, three DOFs on each arm and six DOFs on each leg) with XM-430-350R and an Intel dual-core i3 based NUC (8 GB DDR4 SODIMMs 2,133 MHz and 128 GB M.2 SSD). Each robot uses a sub-controller, OpenCR which is equipped with an IMU that provides a 3-Axis Gyroscope, 3-Axis Accelerometer, and 3-Axis

Magnetometer, and uses U2D2 for actuator communication. Each robot uses a Logitech C920 webcam for visual feedback.

## UNIFIED ROS FRAMEWORK

In previous years, our robots faced the challenge of operating under two separate ROS frameworks. While Oscar utilized the Robotis-OP3 ROS framework from *Robotis (2024b)*, Polaris relied on a custom ROS framework. Although some modules, like vision, could be shared, others required separate development and maintenance, placing a significant workload burden on our teams. To streamline our operations and enhance efficiency, we made a crucial adaptation by migrating Polaris to the Robotis-OP3 ROS framework, creating a Unified ROS Framework as shown in Fig. 3. This Unified ROS Framework which serves as a central hub for various components essential for robot functionality and interactions with four key elements: Interaction, Multi-event, Networking, and Action. The Interaction component focuses on the communication and exchange of information between robots and humans or other robots, ensuring effective human-robot interaction (HRI) by managing inputs and outputs to enable seamless communication. The Multi-event element handles the management of multiple events or tasks that a robot might need to perform, allowing the ROS framework to coordinate and prioritize various activities, ensuring efficient handling of concurrent operations. Networking involves the establishment and maintenance of communication links between different robots or between robots and external systems, ensuring that the robots can exchange data and collaborate in a networked environment to facilitate distributed operations. The Action component is responsible for the execution of tasks and behaviors by the robot, encompassing the control mechanisms and algorithms that enable the robot to perform specific actions, whether they are pre-programmed or dynamically determined based on real-time data. The Unified ROS Framework integrates these elements to provide a cohesive and efficient platform for robotic operations, enabling robots to interact, manage multiple events, network, and perform actions in a synchronized manner.

The Unified ROS Framework not only streamlined our operations but also provided enhanced robustness by supporting complete functionality in both robots. To accommodate the differences in the mechanical structure, actuators, and sub-controllers between Oscar and Polaris, we updated the motion and peripheral modules, allowing Polaris to integrate seamlessly into the Robotis-OP3 ROS framework. Essential modules, such as action and walking modules, were made accessible to Polaris through this transition. Figure 4 presents an illustrative representation of the design of our Unified ROS Framework with some relevant modules that represent a significant advancement in streamlining our robot musicians' operations.

The Unified ROS framework offers an advanced platform for the operation of two uniquely designed robotic musicians, Oscar and Polaris, capable of performing on the keyboard and drum respectively. The architectural configuration of this framework is multifaceted, integrating a range of modules that converge with the central ROS Master, also known as the Parameter Server—a centralized repository for all robot parameters. This server is universally accessible to every module, guaranteeing uniformity and

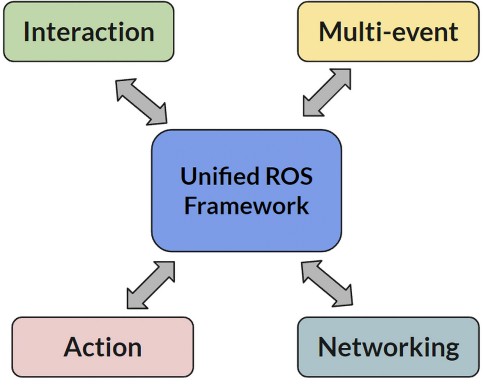

**Figure 3** The unified ROS framework.

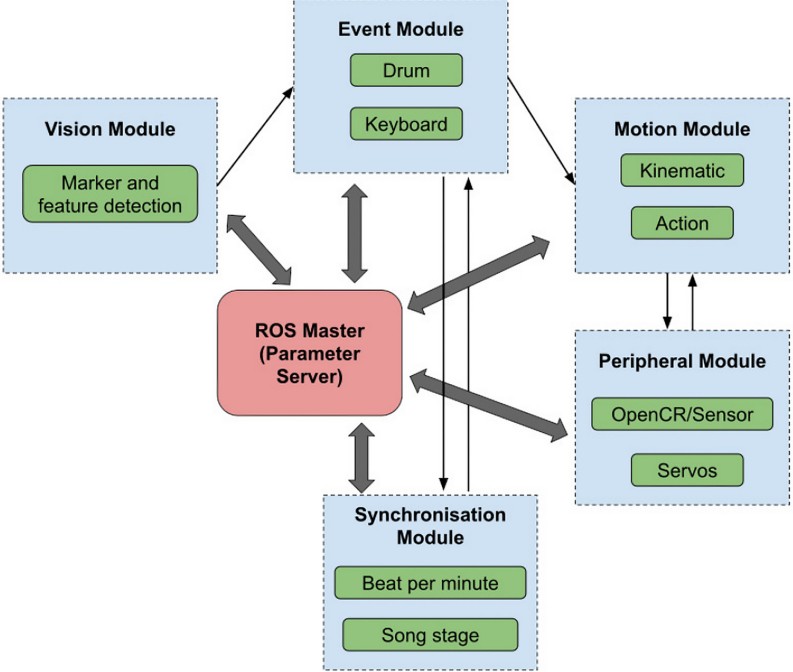

**Figure 4** The unified ROS topics.

consistency in the operational parameters of the robots. The framework consists of multiple modules as shown in Fig. 3. Each module integrates multiple ROS nodes, which act as the core building blocks within the ROS computational graph. For example, the Event module consists of various nodes related to specific musical events like drumming and keyboard playing. This module taps into the Vision module by subscribing to a marker node topic. By doing so, it receives essential data for visual servoing. After processing, the Event module then relays the necessary event data to the Motion module, guiding the robots' physical actions during performances.

When it comes to maintaining rhythm and harmony, the Synchronization module plays an important role. Equipped with BPM (beats per minute) and song stage synchronization tools (*e.g.*, verse and chorus), this module ensures the robots stay harmoniously

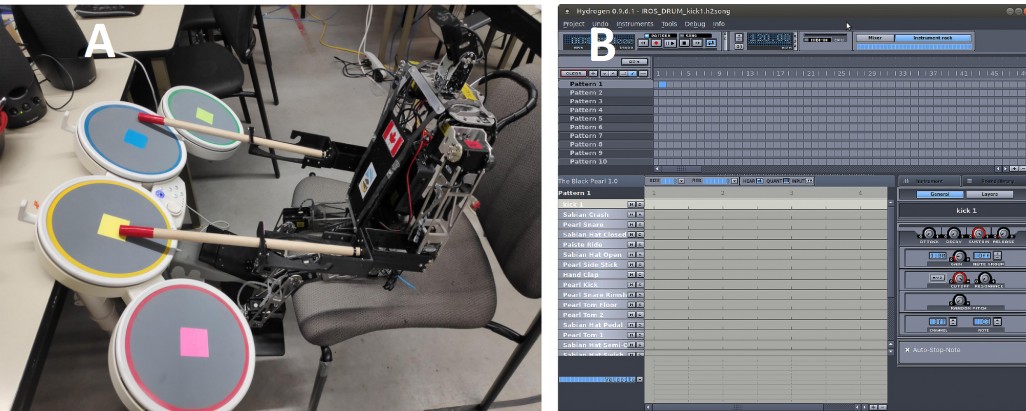

**Figure 5** (A) Drummer polaris and (B) drum MIDI emulator: Hydrogen.

synchronized throughout their performance. Specifically, Oscar, the keyboard-playing robot, acts as a server, broadcasting song stage cues. Polaris, the drum-playing counterpart, operates as a client, interpreting these messages to synchronize its drumming. This dynamic interplay between Oscar and Polaris guarantees reasonable beat synchronization, with Oscar signaling song stage transitions and Polaris adjusting its tempo in real-time which is discussed in the Discussion Section.

## Drummer implementation

Our drummer, Polaris, along with advancements made to improve its robustness and flexibility in playing Wii Rock Band's drum set, is depicted in Fig. 5A. The drum set is interfaced with Polaris's computer *via* the JoyToKey software (or Joy2Key) developed by *joy2key (2024)*, enabling the translation of drum hits into keyboard inputs. This permits midi applications like Hydrogen, illustrated in Fig. 5B, to be operated and mapped using the drum kit. For instance, JoyToKey will translate any hit on a drum or pedal into specific keyboard strokes, tricking the targeted application into believing a real keyboard is in use. This mechanism facilitates the customization of our drums to produce any desired beats and sounds. The combination of kick drum, snare drum, hi-hats, ride cymbal, and crash cymbal has been programmed to enhance the musical versatility and richness of Polaris's drums play.

In terms of drum motion control, Polaris adopts the same strategy as Oscar's visual servoing and inverse kinematic module, both integrated into our Unified ROS framework. Instead of merely replicating pre-determined drum beats, we have enhanced Polaris's performance by incorporating "sheet music" reading abilities. Now, it can dynamically adjust its play according to the sheet music displayed on the screen as shown in Fig. 6. Our implemented vision module captures the sheet content and identifies the notes to play for each frame. These notes then serve as inputs to Polaris's motion control module, as showcased on a program similar to Rock Band. However, this feature can be influenced by inconsistent lighting conditions. In cases where lighting is insufficient, the system reverts to the default drum play, free of visual inputs from the screen.

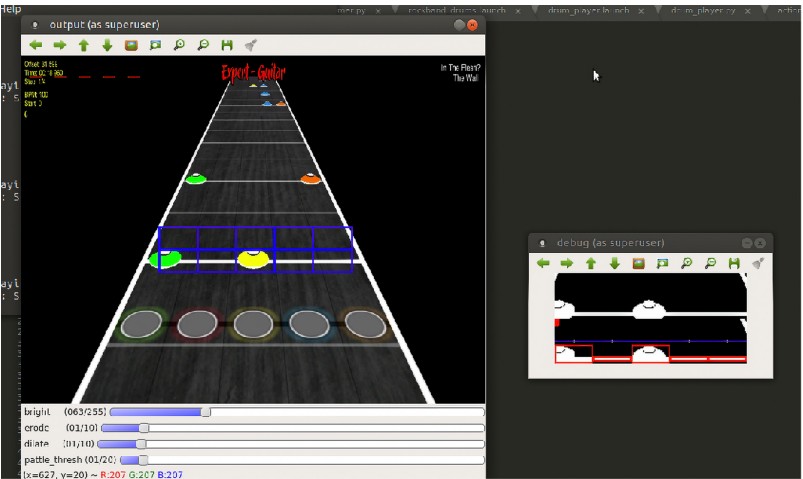

**Figure 6 Rockband-like drum sheet reading.**

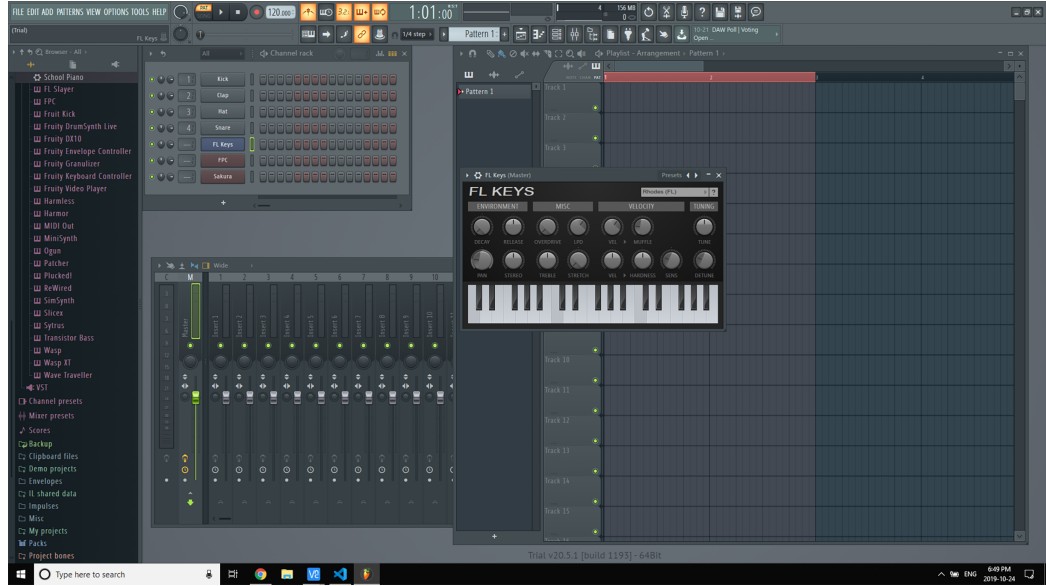

**Figure 7 Keyboard mixers: FL studio.**

## Keyboardist implementation

Our humanoid keyboardist, Oscar, is equipped with the capability to play an Akai MPK Mini Play (https://www.akaipro.com/mpk-mini-play; Akai Professional, Cumberland, RI, USA) or any similar musical keyboard. The Akai MPK Mini Play is a portable MIDI keyboard based on the renowned Akai Professional's MPK Mini, featuring 128 sounds and an integrated speaker. It boasts eight drum pads and a four-way joystick, providing Oscar with the necessary tools to perform notes and chords proficiently. With the addition of four knobs, Oscar can manipulate eight different parameters in real-time, expanding its performance possibilities. For sound effects and loop recording, we utilize FL Studio, as shown in Fig. 7.

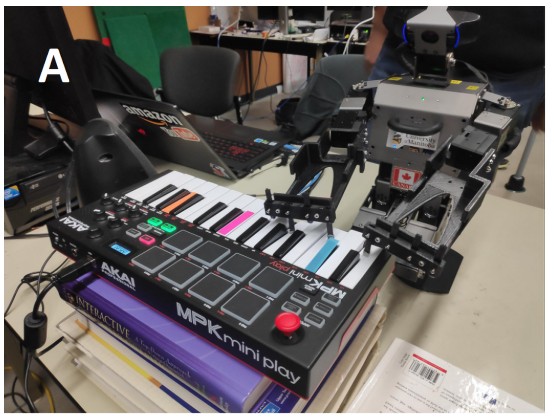
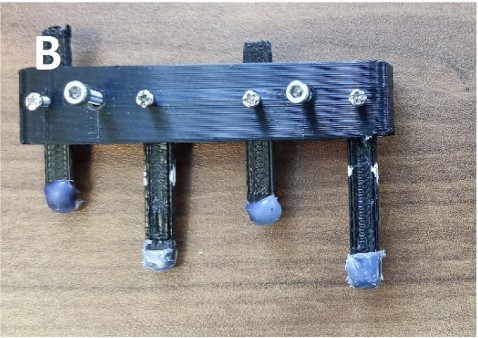

**Figure 8 (A) Keyboardist oscar and (B) customizable fingers.**

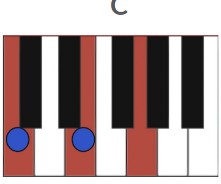
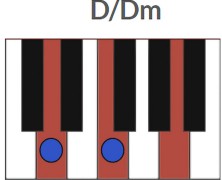
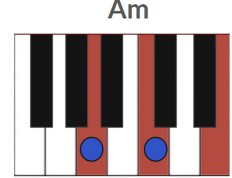
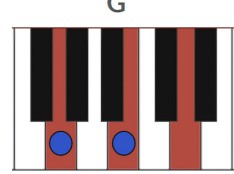

**Figure 9 Simplified chords.**

To orchestrate Oscar's motions during performances, we used the Robotis-OP3 Action Editor to program pre-defined movements for playing all keys on an AKAI MPK Mini Play keyboard. Taking Oscar's customizable fingers into account as shown in Fig. 8B, we designed two playing styles. The first allocates one finger to each hand, enabling the playing of individual white keys, with the left hand handling the bass keys to the left of the central C-key, and the right hand doing the same for the opposite side. The second style employs one finger on the left hand and two on the right to play chords, with selected primary chords predefined, such as C major, A minor, G major, *etc.*, to accommodate finger customization limitations. While this decision was made to accommodate the robot's physical limitations in playing complex chords, we recognize that this can introduce ambiguity from a musical perspective, as bichords can be interpreted as either major or minor chords depending on the context. Although this simplification does not impact the robot's ability to participate in live performances, future iterations of the system could explore ways to enable the robot to play full triads or even more complex chords, enhancing the musical appropriateness of its performance. Figure 8A shows the initial setup of our humanoid keyboardist, Oscar.

Figure 9 shows simplified chord diagrams created to enable the humanoid robot Oscar to play various songs on a keyboard. The example chords shown are C, D/Dm, Am, and G. Each diagram illustrates the keyboard with specific keys highlighted in red, and blue dots indicate the exact keys to be pressed. For example, in the C chord, the diagram highlights the keys representing the notes C and E, with blue dots showing which keys to press to form the chord. The D/Dm chord diagram displays the notes D and F#, with blue dots

marking the keys needed to play either the D major or D minor chord. In the Am chord diagram, the highlighted keys include the notes A and C, and the blue dots indicate the specific keys to press for the A minor chord. Finally, the G chord diagram shows the notes G and B, with blue dots marking the keys to press to form the G chord. These simplified chord diagrams are tailored to Oscar's capabilities, ensuring that the humanoid robot can accurately and efficiently play these chords during musical performances.

To enhance the robustness of Oscar's keyboard playing motions, we implemented a visual servoing (*Chaumette & Hutchinson, 2006*; *Hutchinson, Hager & Corke, 1996*) approach, providing real-time positioning correction during live performances. This approach utilizes two markers: one on the robot's right hand and the other on the middle B-key of the keyboard. Starting from an initial position centered on the keyboard, Oscar calibrates the positions of its hands to optimize the distance. It further calibrates along the x-axis by moving its hand from left to right, based on the x-axis distance. Subsequently, calibration takes place along the y-axis (key-press) by comparing the y and height of each marker to determine the appropriate key-press depth. Throughout the calibration process, the servos make adjustments by one degree to optimize the final calibration, relying on the inverse kinematics module. During the performance, the positioning dynamically corrects itself if required, ensuring precise key reach with visual servoing control which is further discussed in the 'Visual servoing implementation' subsection.

## Visual servoing implementation

Since real-time image processing is required to locate the musical instruments and robot's own hands to facilitate accurate interaction with the instrument. A color-based image visual servoing approach is implemented as mentioned in previous sections and our previous work (*Lau, Anderson & Baltes, 2019*). This approach aims to segment the colored markers on the instruments and the robot's hands from the background, thus enabling the robot to determine their positions and movements accurately. By resizing camera feedback to 480 × 360 pixels, we enhance the processing speed without significantly compromising the visual information essential for the task at hand; a critical requirement for visual servoing applications where the robot must respond to dynamic visual information promptly. We employed a color segmentation algorithm that differentiates between the color profiles of the piano, the robot's hands, and the background. For example, as shown in Fig. 10, the colors present in the piano keys and the robot's hands are identified and isolated from the rest of the image using a color space that is less sensitive to illumination changes, such as HSV (Hue, Saturation, Value). After segmentation, morphological operations are applied to remove noise and fill gaps within the segmented regions to ensure a consistent outline of the piano and hands. Using the segmented images, the system calculates the position and orientation of the piano keys relative to the robot's hands. This information is fed into a control loop that adjusts the robot's arm and hand movements to align with the keys, allowing for the accurate striking of notes. However, the efficacy of the color segmentation process depends heavily on consistent lighting conditions and the distinct color profiles of the objects of interest. Under variable lighting, the system may require dynamic thresholding techniques to maintain segmentation accuracy.

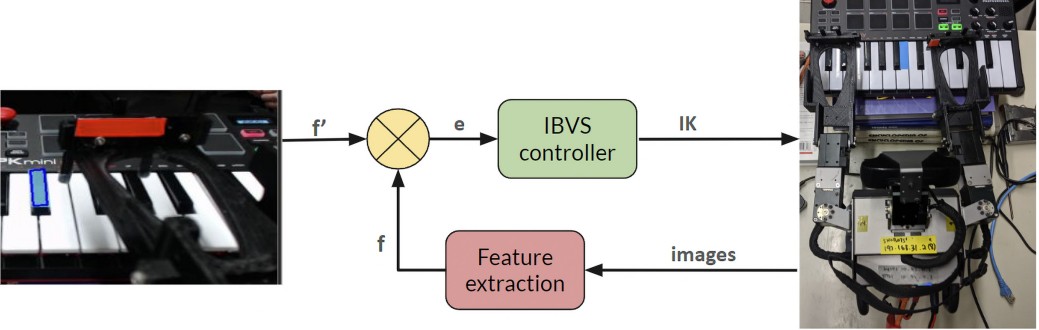

**Figure 10 Oscar's piano visual servoing controller.**

Figure 10 illustrates the visual servoing controller system used by the humanoid robot Oscar to play the keyboard. The setup demonstrates how visual feedback is employed to adjust the robot's hand movements to accurately press the keys. On the left side of the figure, a close-up of the keyboard shows the C key marked with a blue indicator and the robot's hand marked with a red indicator. This visual information serves as the feedback mechanism for the controller. The process begins with the Feature Extraction stage, where images captured by the camera are processed to identify key features, such as the blue marker on the C key and the red marker on the robot's hand. This extracted feature information, denoted as $f$, is sent to the IBVS (Image-Based Visual Servoing) controller. The IBVS controller is at the core of this system. It compares the current feature information $f$ with the desired feature $f'$. Based on this comparison, it calculates the error $e$, which represents the difference between the current and desired positions of the robot's hand relative to the keyboard keys. The error $e$ is then used to generate the appropriate control commands through inverse kinematics (IK). These commands are sent to the robot's actuators to adjust the hand's position and movement. This closed-loop system ensures that the robot's hand accurately moves to press the correct keys on the keyboard. On the right side of the figure, a wider view of the setup shows the entire keyboard with the robot's hands positioned to play. This visual servoing controller system, by continuously processing visual feedback and adjusting the robot's hand movements, enables Oscar to play the keyboard accurately and efficiently.

In this study, we implemented a color-based image visual servoing technique, which allows real-time tracking of the robot's hands and the musical instrument using a monocular camera. While this approach simplifies the visual feedback process by focusing on color markers, we acknowledge that it does not inherently provide depth information. To overcome this limitation, the visual servoing system dynamically adjusts hand positions along the z-axis based on the estimated distance between the robot and the instrument. Depth perception, while crucial for musical performance, is approximated through calibration and real-time feedback adjustments. In future iterations, we plan to incorporate stereo vision or depth sensors to improve accuracy. Our technique offers robustness in environments with well-controlled lighting and distinct color markers, though further

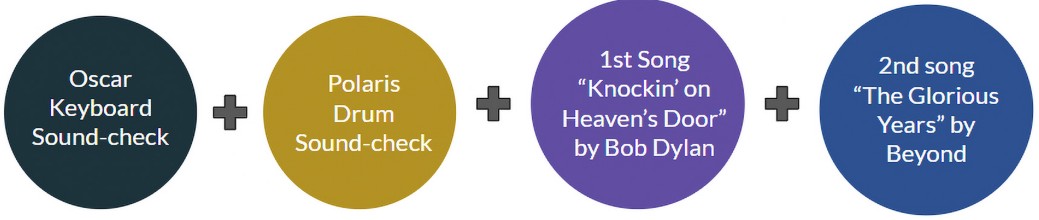

**Figure 11 Collective performance in IROS HAC 2019.**

experimental results are needed to compare its performance with more sophisticated systems like WABOT-2.

## Collective performance

Figure 11 provides a breakdown of a collective musical performance that integrates both human and robotic musicians. This collective performance is further enriched by the dynamic interactions between the MC, Chris (the human DJ), and the robotic musicians, Oscar and Polaris as shown in Appendix A. The sequence begins with a sound-check for Oscar. This initial step ensures that all the equipment and settings for the keyboard are properly configured, allowing Oscar to perform at optimal levels. Sound-checks are crucial in any performance to prevent technical issues and ensure that the sound quality is clear and balanced. Following Oscar's sound-check, the attention shifts to Polaris. Polaris requires a sound-check to align its drumming patterns with the overall rhythm of the performance. This step is vital as it ensures that the robotic drummer's timing and sound output are perfectly synchronized with the human musicians, creating a cohesive musical experience.

The performance kicks off with the first song, "Knockin' on Heaven's Door" by Bob Dylan. This classic piece serves as the opening number, featuring a collaboration between Oscar on the keyboard, Polaris on the drums, a human guitarist, and a human vocalist. The second song in the performance is "The Glorious Years" by Beyond. Continuing the theme of human-robot collaboration, this song further demonstrates the seamless interaction between the human guitarist and vocalist with the robotic drummer and keyboardist. The performance of these songs underscores the potential of humanoid robots in contributing to complex musical pieces, offering a new dimension to live performances.

A significant challenge in this performance is maintaining precise timing, especially during beat transitions. Beat transitions are critical moments in any musical piece, requiring exact synchronization to ensure the music flows smoothly. The robotic drummer, Polaris, must adjust to these transitions with the same dexterity and responsiveness as a human drummer. This challenge is addressed through advanced programming and real-time adjustments, ensuring that the timing remains flawless throughout the performance as described in the Discussion section.

Overall, this collective performance showcases a seamless integration of human-robot interaction in music. It underscores the potential for future collaborations that extend

beyond traditional musical boundaries, providing audiences with a unique and innovative experience.

## DISCUSSION: CHALLENGE AND EXPERIMENTAL RESULTS

The primary component of our presentation consists of a collective performance by our robots, Polaris and Oscar, in coordination with a human guitarist and vocalist. Along with a pre-programmed musical sequence, both robot musicians exhibited the capacity for independent operation, similar to the traditional methods of synchronization in robotic systems, such as those used in Shimon and WABOT-2. For the realization of a comprehensive musical piece, we facilitated a full-length collaboration between the robotic musicians and human performers, introducing a compelling element of human-robot interaction. Despite earlier segments outlining the establishment of musically appropriate movements conducive to performance, a significant challenge was encountered involving maintaining precise performance timing, including the transition from the execution of one beat to the subsequent one. Given the mechanical delays inherent in the servos and network communication, especially over Wi-Fi, achieving real-time synchronization was critical to the success of the performance.

Our initial approach employed a simplistic methodology whereby each robot executed its movements in synchrony, under the assumption that the predetermined tempo (measured in beats per minute, BPM) of the piece would ensure a sufficient degree of uniformity. The primary goal of this method was to synchronize timing by initiating each robot's operations simultaneously. However, this approach was found to be insufficient due to inherent latencies such as those encountered in the servos' TTL and network communication. As a result, maintaining consistent timing synchronization for each robotic entity throughout the entire musical piece proved to be a challenge (*Chakraborty & Timoney, 2020*). For instance, if the drummer robot had just engaged the snare, the initial location of the drumstick for the subsequent beat would deviate which significantly affects the timing of the overall performance. The effect of missed synchronization could be seen in the experiment where Polaris constantly experienced a delay due to its movement latency. This latency caused noticeable discrepancies in the coordination between Polaris, Oscar, and the human performers, disrupting the intended harmony and rhythmic flow of the piece. The delay not only impacted Polaris's ability to keep pace with the pre-programmed sequence but also affected the seamless integration of its contributions within the ensemble, leading to a disjointed musical experience.

To address the inadequacies of the initial method, we transitioned to an active BPM synchronization approach, leading to a substantial enhancement in the performance of the play. In this updated approach, each robot utilizes the uniform tempo of the musical piece to time movements and achieves synchronization *via* a UDP message, systematically orchestrated at each stage (such as intro, verse, chorus, *etc.*) of the musical piece by a server —Oscar, in this instance. Utilizing UDP messaging allowed both robots to execute the musical piece in a synchronized and time-precise manner throughout the entire duration.

In our current implementation, the interaction between the operator and the robots is limited to Oscar, the keyboard-playing robot, while Polaris, the drummer, synchronizes to

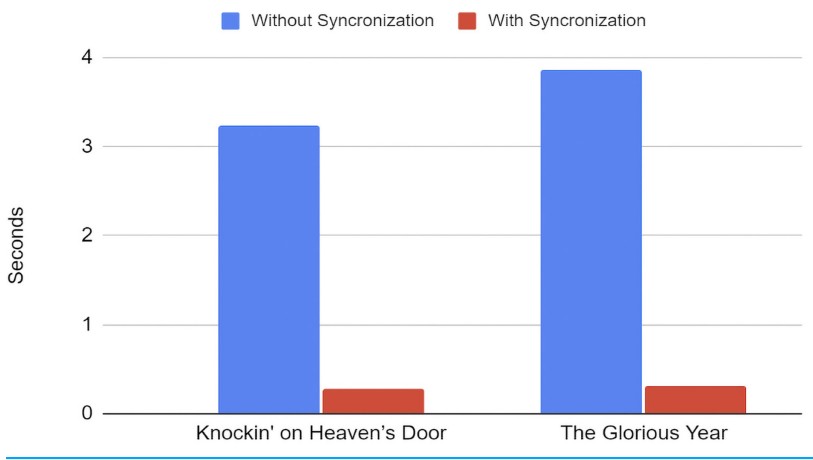

**Figure 12 Offsets timing improvement.**

Oscar's tempo cues. This design choice is primarily due to the distinct roles of each robot in the musical performance. Oscar, being the lead instrument, requires direct input from the operator to initiate and control the musical sequence, whereas Polaris follows the cues provided by Oscar to ensure rhythmic synchronization. Extending operator interaction to both robots could introduce additional complexity in maintaining tempo synchronization. However, future iterations of this system could explore enabling interaction with both robots, allowing for greater flexibility and enhancing the collaborative capabilities of the system.

In order to validate our synchronization approach, we conducted experiments on the timing performance under two distinct setups: (1) Without UDP message passing synchronization, and (2) with UDP message passing synchronization. The test was performed on two separate musical pieces; namely, "*Knockin' on Heaven's Door*" by Bob Dylan and "*The Glorious Years*" by Beyond. Our findings revealed that in the absence of UDP message passing synchronization, the timing variance between Polaris and Oscar was off by an average of 3.55 s or at the rate of 1.87% at the conclusion of both songs as shown in Fig. 12. This deviation was primarily attributed to differential local latencies in the servos and network communication of each robot. On the other hand, incorporating UDP message passing synchronization considerably minimized this timing gap, maintaining an average offset in timing at 0.3 s or at the rate of 0.16% as shown in Table 1. Offset rates are calculated based on the division of musical piece length and their respective offsets. The residual minor variance in timing could be traced back to network latency between the server (Oscar) and client (Polaris). Despite these minor delays, the active UDP message passing substantially enhanced the overall timing coordination between the two robots. By winning 2nd place in both 2019 and 2020 Humanoid Application Challenge competitions, it is clear how enjoyable our robot musicians are perceived and received by human judges and audience. The t-test results provide further insight into the impact of synchronization on timing offsets for each song. For "Knockin' on Heaven's Door," the paired t-test between the offset times with and without synchronization yields a t-statistic with a $p$-value of approximately $1.66 \times 10^{-21}$. This extremely low $p$-value confirms that

**Table 1 Comparison of offsets timing.**

| Musical piece | Length, s | Offset, s and % | | | |
| --- | --- | --- | --- | --- | --- |
| | | Without sync. | Rate | With sync. | Rate |
| Knockin' on Heaven's Door | 175 | 3.63 | 1.97 | 0.28 | 0.16 |
| The glorious years | 205 | 3.45 | 1.77 | 0.31 | 0.15 |
| Average | 190 | 3.54 | 1.87 | 0.295 | 0.16 |

the observed difference in offset times is statistically significant, indicating that synchronization effectively reduces timing discrepancies for this song. Similarly, for "The Glorious Years," the paired t-test results in a $p$-value of approximately $2.55 \times 10^{-24}$, which also strongly supports the significance of the synchronization effect. Both tests show that synchronization produces a highly reliable improvement in timing accuracy, with "The Glorious Years" showing a slightly more significant reduction in offset. These findings reinforce that synchronization has a substantial, statistically significant effect on improving performance for both songs, though the degree of improvement varies slightly with each song.

In addition to qualitative insights and quantitative timing synchronization of the performance, we conducted an experiment to evaluate the robustness and adaptability of the visual servoing system, particularly Oscar's keyboard playing. This experiment involves testing the Oscar's ability to locate and press a color-marked key (C-key) on a keyboard moved to six distinct offset positions. Each offset represents a slight displacement of 1 cm, simulating the distance between adjacent keys. This setup aims to test the adaptability of the robot's visual servoing system by examining its accuracy and responsiveness when the keyboard is repositioned. The experimental setup places the keyboard on a movable platform that allows precise 1 cm adjustments in all six directions covering all x, y, and z axes.

During the experiment, the keyboard is moved to each of the six offset positions randomly with 20 trials for each offset positions to the total of 120 trials for all axes. These offsets are denoted with x_n, x_p, y_n, y_p, z_n, and z_p where _n and _p represent −1 cm and +1 cm respectively. At each new offset position, the robot's visual servoing system detects the color-marked C-key and adjusts its hand position to press the target. For each trial, Oscar must capture the position of the marker and adjust its hand accordingly to match the offset. This process repeats across all six offsets to evaluate Oscar's ability to handle changes in the x, y, and z axes.

The experimental results in Table 2 and Fig. 13 show varying success rates across different offset positions, Offsets y_n and z_p have perfect success rates (100%), resulting in highly significant differences from the overall mean success rate (94.2%, $p < 0.001$). These high accuracy observed for Offsets y_n and z_p can be attributed to the displacement of 1 cm being relatively small for the key and motion. As a result, the robot can effectively compensate for this shift without experiencing significant errors, leading to a perfect success rate. For the other offsets, the differences are not statistically significant

**Table 2  Success rates and keypress outcomes for each offset position.**

| Offset position | x_n | x_p | y_n | y_p | z_n | z_p |
|---|---|---|---|---|---|---|
| Correct keypress | 18 | 19 | 20 | 18 | 18 | 20 |
| False keypress | 2 | 1 | 0 | 2 | 2 | 0 |
| Success rate (%) | 90 | 95 | 100 | 90 | 90 | 100 |

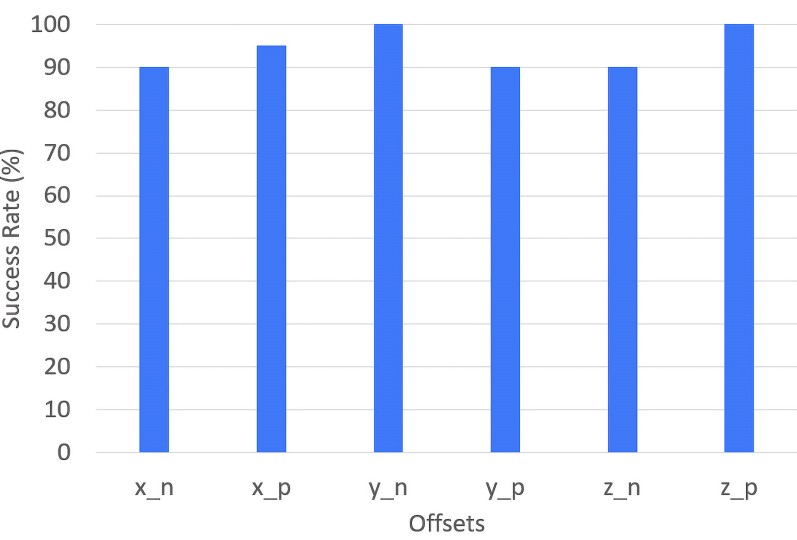

**Figure 13  Visual servoing offsets success rate.** 

($p > 0.05$), indicating that their success rates (90% or 95%) are reasonably close to the mean. Overall, these results suggest that while performance varies slightly by direction, the system's adaptability is consistent across offsets within these results.

## CONCLUSIONS

This article discusses the comprehensive effort and work put into preparation for the Humanoid Application Challenge (HAC) competition by creating a robot-human musical band. The primary focus of our work centers on the orchestration of collaborative performance, integrating both humanoid and human band members, whilst ensuring precise synchronization of musical timing. Our research presents a thorough analysis of the musical performance, in addition to the corresponding hardware and software specifications. As we venture into the future, our goal is to augment our research in a way that improves the reliability and entertainment value of our robotic musicians during live performances. Another critical aspect of this enhancement is refining our synchronization methodologies through optimized offset timing, thereby ensuring a seamless and harmonious performance. One of the approaches is musical improvisation as shown by *Weinberg & Driscoll (2007)*, *Hoffman & Weinberg (2011)*, which relies on heuristics that could predict the offsets of the musical piece based on the delay of a few initial musical stages. The other approach as proposed by *Mizumoto et al. (2008)* is to allow the robots to recognize beat structures, express these beats, and suppress self-generated counting sounds

to improve recognition accuracy. Another key objective of our future research is to enhance the social interaction capabilities of our robots. In this ambitious pursuit, we aim to equip our robots with the ability to interpret musical notes and engage meaningfully with human audiences. We believe music, as a powerful form of entertainment, can spark and deepen public interest in humanoid robots. Through this work, we hope to make significant contributions to advancing the field of humanoid robotics.

## APPENDIX A
## COLLECTIVE PERFORMANCE SCRIPTS

| | |
|---|---|
| **MC** | Introduce the band members: Chris, Oscar, and Polaris. |
| **Action** | Oscar and Polaris should wave or play a key. |
| **MC** | Hello! Snobots-Band is ready to rock! Let's do a sound check. |
| **MC** | Chris, how is our keyboardist Oscar doing? |
| **Chris** | Oscar is a little shy, hey Oscar, are you ready? |
| **Oscar** | No. |
| **Action** | Oscar shakes his head. |
| **Action** | Oscar performs calibration then enters into ready mode. |
| **Chris** | Okay, what popular song do you know, Oscar? |
| **Oscar** | I know Michael Jackson. |
| **Chris** | Does anybody here know Michael Jackson? (to the audience) |
| **Audience** | YES. |
| **Chris** | Oscar, do you think you could play Billie Jean? |
| **Oscar** | I think I can. |
| **Action** | Oscar plays Billie Jean. |
| **MC** | What about Polaris? |
| **Action** | Polaris performs calibration then enters into ready mode. |
| **MC** | Okay, good. Can you read music and play at the same time, Polaris? |
| **Polaris** | Of course. |
| **Action** | Polaris nods his head. |
| **MC** | Let's do it! |
| **Action** | Polaris plays Rockband. |
| **MC** | Sounds good. But can we play together as a band? |
| **Polaris** | Yes. |
| **MC** | Let's play Knockin' on Heaven's Door! |
| **Action** | Polaris, Oscar, and MC play Knockin' on Heaven's Door. |
| **Chris** | That was awesome! But MC, we are in Macao. Do you know any song in Cantonese? |
| **MC** | Yes Chris, we can play a Cantonese song! |
| **Action** | Polaris, Oscar, and MC play The Glorious Years. |
| **MC** | Thank you! |

### Funding

This work was supported by the Department of Computer Science, University of Manitoba and School of Engineering and Computer Science, Laurentian University. The funders had no role in study design, data collection and analysis, decision to publish, or preparation of the manuscript.

### Grant Disclosures

The following grant information was disclosed by the authors:
Department of Computer Science, University of Manitoba.
School of Engineering and Computer Science, Laurentian University.

### Competing Interests

The authors declare that they have no competing interests.

### Author Contributions

- MengCheng Lau conceived and designed the experiments, performed the experiments, analyzed the data, performed the computation work, prepared figures and/or tables, authored or reviewed drafts of the article, and approved the final draft.
- John Anderson analyzed the data, authored or reviewed drafts of the article, and approved the final draft.
- Jacky Baltes analyzed the data, performed the computation work, authored or reviewed drafts of the article, and approved the final draft.

### Data Availability

The raw data is available in the Supplemental Files.

### Supplemental Information

Supplemental information for this article can be found online at http://dx.doi.org/10.7717/peerj-cs.2632#supplemental-information.

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
