# Peer review of "Integrating humanoid robots with human musicians for synchronized musical performances"

_PeerJ Computer Science, doi:10.7717/peerj-cs.2632_

## Round 0.1 · original submission · Major Revisions

Dear authors,

You are advised to critically respond to all comments point by point when preparing an updated version of the manuscript and while preparing for the rebuttal letter. Please address all comments/suggestions provided by reviewers, considering that these should be added to the new version of the manuscript.

Kind regards,
PCoelho

Reviewer 1 ·

Basic reporting

-Lines 204-206 are presented earlier than the figure (line 213). Despite being used as an example, this example requires further information, which are provided later in the text, to be functional.
-Line 152 states that Figure 6a presents the initial setup of Oscar and its viewpoint; however, its viewpoint is not presented.

Experimental design

no comment

Validity of the findings

The novelty of the manuscript should be better presented. Despite the framework for the humanoid robot band has not been presented in previous works, it is difficult to understand the novelty of the manuscript. For example, the introduction does not present the unified framework nor the BPM synchronization used to improve the system's performance. Moreover, the manuscript seems more like an application of a previous framework.

Additional comments

If this reviewer understands correctly, the interaction with the operator concerns the keyboard-playing robot, while the other robot synchronizes to the former. Is there any reason to not make both robots interact with the user?

Reviewer 2 ·

Basic reporting

The paper is written in correct English and easy to read.
Literature review needs improvement to cover related studies and to clarify the objective of this study.
These include existing keyboard playing humanoid (e.g., WABOT-2), drum playing humanoid (e.g. HRP2), image-based recognition of music instruments and music scores,
and synchronization techniques between robots and humans.

Experimental design

The paper does not clearly state concrete research questions.
Quantitative evaluation of the developed systems was not presented apart from evaluation of timing synchronization.

Validity of the findings

Discussion made is mostly qualitative and not founded by experimental data.

Additional comments

This paper reports the development of a robotic system consisting of two humanoid robots, one capable of playing a keyboard and another a drum set, that can perform music with human musicians.
Main technical elements are tracking of relative position of the musical instrument and the robot by means of visual servoing and timing synchronization between robots by UDP communication.
Live performance was made during the Humanoid Application Challenge.

I respect the authors' interest towards entertainment robotics and their effort to build a robotic system that can play real musical instruments. The paper, however, is mostly dedicated to the description of the system developed, and its contribution from an academic perspective is weak.

In is not clear if the visual servoing technique used in this study is any better than ones used in past studies. For example, the famous WABOT-2 robot could play a keyboard while reading music scores with cameras. The need for colored markers could be disadvantageous in some cases.
No exprimental results were shown concerning the performance of visual servoing in terms of its precision and robustness. It was also unclear how depth information, which seems crucial for play musical instruments, was measured using a monocular camera.

The paper shows comparison of the drift of timing between the two robots when the UDP-based synchronization was used or not used. Timing synchronization by means of network communication sounds reasonable, but it rather seems to be a common practice. Moreover, the robots cannot synchronize with human music performance in this manner.

The authors claim that "seamless integration of human robot interaction" was realized, but it sounds overstated. There seems to be no mechanism in the robots' side that enable them to synchronize with human musicians. They play music in their own preprogrammed timing, and human players follow their pace.

·

Basic reporting

This paper gives an overview of a human-robot band as developed for the Humanoid Application Challenge. Overall it is easy to read with a large amount of description of how the robots were developed and adapted to become a keyboard player and drummer.

The introduction gives a clear background with current musical robot projects as a context for this work. However, missing from this contextual opening is what is the Humanoid Application Challenge, who organizing it, and what its aims are. This needs to be addressed so that the reader can understand the level of significance of this award.

Experimental design

The main issue with this paper is that starts as a report about a human-robot band, but ends up being about research specific to challenges with synchronization and timing when playing music with robots. The problem is not introduced in the beginning of the paper but somehow emerges later as a research problem while making the robots. Therefore this paper does not have a well defined research question and it is unclear if the research presented is meaningful.

Overall this paper needs to be reworked to clearly identify the research problem to which the data speaks to and how this is important to the field of robot music.

Validity of the findings

The findings of using the UDP protocol are clearly compared to the previous communication protocols. However it feels like the quantitative research was added after the fact to make the research some how more valid, rather than addressing the issue of time within the design discussion earlier in the paper, which mostly deals with how notes are played by the robots.

Reviewer 4 ·

Basic reporting

Language is clear and unambiguous. There are a few typos, I encourage authors to read it carefully.
Strangely, in-text references are reported without brackets.

Literature background is sufficiently described.

The structure of the paper is straightforward and clean.

All relevant terms/concepts are described.

Experimental design

The experiment is only part of the paper's message, which presents two musical robots as the outcome of the design and development process. These robots are then tested in a live context during a conference.

Novelties are well highlighted.

Methods are described in sufficient detail to replicate.

Validity of the findings

Findings are more related to the successful development and integration of the robots, rather than with the actual results of the synchrony/live experiment.

Additional comments

I have some concerns with the reduction of major triads (e.g., CEG) to bichords (e.g., CE). These might be perceived as ambiguous (i.e., either as major triad CEG or as minor triad ACE). I'm not sure this is relevant from a robotic/computer science perspective, but it is surely so for the musical relevance/appropriateness of the paper.

Watching the video, I noticed that robots require some adjustments of the starting position concerning the musical instrument/setup. Wouldn't a more constrained setup (e.g., mechanical linkage between robot and instrument) facilitate the whole session?

---

## Round 0.2 · Major Revisions

Dear authors,

You are advised to critically respond to all comments point by point when preparing an updated version of the manuscript and while preparing for the rebuttal letter.

Some quantitative results and evaluation of the performance of the framework should be added.

Please address all comments/suggestions provided by reviewers, considering that these should be added to the new version of the manuscript.

Kind regards,
PCoelho

Reviewer 1 ·

Basic reporting

no comment

Experimental design

no comment

Validity of the findings

The manuscript fails in presenting a quantitative analysis that would improve its scientific soundness. The main result of the manuscript seems the effectiveness of the presented framework, but the metrics provide to evaluate the performance of the framework are insufficient for a proper quantitative analysis. Indeed, the discussion is mostly qualitative and not based on any experimental data, and this reviewer believes that it is required for the manuscript and not for future works.

Reviewer 2 ·

Basic reporting

no comment

Experimental design

no comment

Validity of the findings

no comment

Additional comments

The authors applied drastic revision to the paper, and now it seems to be in much better shape.
Specifically, in the introduction, some additional background of HAC is given, and it is made clear that
the main conceptual novelty is in realizing music performance using off-the-shelf humanoid robots
in contrast to using dedicated hardware, and that the main technical contribution is in timing synchronization between the robots.

Although some of the critical comments raised in the previous round are not directly dealt with in the paper,
sufficient discussion is made in the paper justifying limitations of the developed system
and pointing out these problems are left for future study.

Reviewer 4 ·

Basic reporting

The article improved with respect to the minor issues raised in my previous review.

Experimental design

No comment

Validity of the findings

No comment

---

## Round 0.3 · accepted · Accept

Dear authors, we are pleased to verify that you meet the reviewer's valuable feedback to improve your research.

Thank you for considering PeerJ Computer Science and submitting your work.

Reviewer 1 ·

Basic reporting

no comment

Experimental design

no comment

Validity of the findings

no comment

Additional comments

The authors successfully answered all suggestions.